# The Influence of Oxidant on Gelatin–Tannin Hydrogel Properties and Structure for Potential Biomedical Application

**DOI:** 10.3390/polym14010150

**Published:** 2021-12-31

**Authors:** Konstantin Osetrov, Mayya Uspenskaya, Vera Sitnikova

**Affiliations:** Bioengineering Institute, ITMO University, 197101 Saint-Petersburg, Russia; mv_uspenskaya@itmo.ru (M.U.); info@BioEngineering.institute (V.S.)

**Keywords:** hydrogel, tannin, gelatin, thermal analysis, sorption, crosslinking density

## Abstract

Nowadays, there is a widespread usage of sodium periodate as an oxidant for synthesizing gelatin–tannin hydrogels. The impact of iodine compounds could have a harmful effect on human health. The study focuses on the proposal of alternative oxidizing systems for tannin oxidation. Gelatin–tannin hydrogels were obtained based on the usage of H_2_O_2_/DMSO/KMnO_4_/KIO_4_ oxidants and characterized with sorption, thermal (TGA, DTG, DSC), mechanical, FTIR and other methods. The sorption experiments were carried out in a phosphate buffer (pH = 5.8/7.4/9) and distilled water and were investigated with Fick’s law and pseudosecond order equation. The pH dependence of materials in acid media indicates the possibility of further usage as stimuli-responsive systems for drug delivery. Thermal transitions demonstrate the variation of structure with melting (306 ÷ 319 °C) and glass transition temperatures (261 ÷ 301 °C). The activation energy of water evaporation was calculated by isoconversional methods (Kissinger–Akahira–Sunose, Flynn–Wall–Ozawa) ranging from 4 ÷ 18 to 14 ÷ 38 kJ/mole and model-fitting (Coats–Redfern, Kennedy–Clark) methods at 24.7 ÷ 45.3 kJ/mole, indicating the smooth growth of values with extent of conversion. The network parameters of the hydrogels were established by modified Flory–Rehner and rubber elasticity theories, which demonstrated differences in values (5.96 ÷ 21.27·10^−3^ mol/cm^3^), suggesting the limitations of theories. The sorption capacity, tensile strength and permeability for water/oxygen indicate that these materials may find their application in field of biomaterials.

## 1. Introduction

Hydrogels are one of the perspective classes among all biomaterials. These are polymeric structures, which have properties similar to soft human tissues [1]. They are highly hydrated, elastically deformable and water and oxygen permeable [2]. These properties could represent a skin-like structure, allowing them to be a functional analogue of skin grafting [3]. Severities considering the negative sides of hydrogels include their poor physico-mechanical properties, the potential toxicity of crosslinking agents, inability to provide antibacterial activity and allergic rejection by the human body [4,5]. The main structural element of hydrogels is monomers. The crosslinking of monomers provides network formation. There are several mechanisms of crosslinking.

Chemical crosslinking provides mechanical strength but potentially uses compounds that may be toxic. Some of them may have a potential harmful effect on humans, so they regarded to be washed from hydrogel network. Widespread crosslinkers include glutaral aldehyde [6], glyoxal [7], genipin [8], etc. Although many of them have a considerable toxic impact on human cells, one of the biggest challenges for a researcher is to choose the appropriate one [9]. Others appear to be non-toxic, but their usage is limited to economical purposes or synthesis difficulties [10].

Among the promising crosslinking agents is tannin, a group of plant-derived polyphenolic components, which are recognized to be safe and appear to offer protection from some bacterial cultures [11,12]. Gelatin has a relative structure as the main component of human skin (collagen). It has a similar amino-acid composition and can form close conformations [13]. Furthermore, gelatin for medical purposes, apart from collagen, is sterile, which allowed its usage to expand to wound dressings. Tannin forms precipitates with peptide molecules if mixed directly in a solution without adjusting the pH level [14]. The pH of tannin solution can be increased from an acidic to moderate basic level (from 4.5 ÷ 5.5 to 9 ÷ 10) to prevent this effect. Furthermore, it provides a medium for easier tannin oxidation and causes the ionization of oxidizer molecules. The formation of a chemically crosslinked polymer network, based on tannin and gelatin, requires the preliminary oxidation of tannin molecules.

Periodate compounds are commonly used for the oxidation of functional groups [15] and could be used for the selective oxidation of vicinal diols to aldehyde groups [16]. Usage of those oxidizers is well known for obtaining hydrogels, linking tannin in their network [17,18,19]. Additionally, increasing the pH in oxidation reactions with periodate-containing substances is another important thing to mention. The formation of an ether complex under iodate prevents the cleavage of C–C bonds and may cause elementary iode from forming [20]. This could be toxic for some biomedical applications.

In our study, we present alternative oxidizing systems for the synthesis of gelatin–tannin hydrogels. Large and hard tannin molecules contain a significant number of hard-to-reach hydroxyl groups. These can be less accessible for wide-radius molecules of oxidizers, which are blocked by a high-density net of hydrogen bonds [21]. Additionally, it is important to note the existence of hard molecule substrates and the sterically hindered effects related to them. In this way, relatively small and more flexible oxidative agents (DMSO and H_2_O_2_) could have better potential for the effective oxidation of tannin molecules compared to KIO_4_ and KMnO_4_. The formation of the hydrogel network is followed by a complex reaction (Figure 1), according to [22]:

In the first stage (ionization), the deprotonation of tannin causes the formation of accessible sites for electron donor molecules of oxidizers. The constants of equilibrium in water media (25 °C) are 11.65 for H_2_O_2_, 35.1 for DMSO and 1.64 for HIO_4_ [23]. This results in different capacities for forming ionizable substrates and the efficacy of the further conversion of hydroxyl groups to aldehyde. Various functionality products of tannin are formed depending on the type of oxidation agent in the second stage (oxidation). In next stage (attachment), oxidized tannin reacts with amine groups of gelatin through a Michael-type reaction or the formation of a Schiff base. The reactions were previously confirmed by IR, UV and HMR spectroscopy [22] on the model reaction of tris-hydroxybenzaldehyde and gelatin. Then, molecules of tannin linked to gelatin starts forming intermolecular bonds and causes the formation of a crosslinked polymer network (network formation).

The purpose of our study was to demonstrate the usability of various oxidizing systems for synthesizing tannin hydrogels. In the following sections, the hydrogels are indicated by the following notations regarding to the usage of proper oxidizing system—H-H_2_O_2_, H-DMSO, H-KMnO_4_ and H-KIO_4_. The prepared hydrogels were investigated to evaluate the influence of oxidizer choice on performance properties and structure. The synthesized materials could be considered as model systems for subsequent research on biomaterials.

## 2. Materials and Methods

### 2.1. Materials

Food-grade substances (gelatin, tannin) were used (supplied by Neva-reaktiv, Saint-Petersburg, Russia). The oxidizers (H_2_O_2_, DMSO, KMnO_4_) were pharmaceutical grade, and KIO_4_ was analytical grade (supplied by Reahim, Moscow, Russia). Every component of buffer salines (K_2_HPO_4_; KH_2_PO_4_; NaCl) and NaOH were analytical grade (supplied by Neva-reaktiv). All substances used in oxygen determination experiments (MnSO_4_, KI, Na_2_S_2_O_3_, starch) were analytical grade (supplied by Neva-reaktiv). All reagents were (and used as received.

### 2.2. Synthesis

A total of 3.08 g of the gelatin (Mr = 40,000 ÷ 100,000 g/mol) was left to freely swell in 0.1% *w*/*w* NaCl buffer saline for 2 h. The solution of gelatin (8% *w*/*w*) was obtained through heat treatment (80 °C) with constant stirring until complete dissolution. The base solution of 0.6 mole oxidizer (3% *w*/*w*, pH = 10) was added to 0.26 g of tannin and heated (80 °C) for 2 h to complete oxidation of tannin. The solutions’ pH was established by adding an appropriate amount of NaOH solution (2 M) and controlled by pH meter (PCE-228, PCE Instruments, Meschede, Germany). After that gelatin and oxidized tannin solutions were vigorously mixed together and heated for 1 h (80 °C) under constant stirring. Finally, homogenous solution was casted in Petri dishes (Ø 90 mm) to obtain a depth of about 8 mm and left to fully crosslink overnight.

All samples were dried to the constant weight under NTP conditions before further investigations, unless stated otherwise. Thickness was measured by micrometer (MDC-1 MJC 293-330, Mitutoyo, Kawasaki, Japan). Linear scale length dimensions were obtained with the callipers (ShC-I 0-150, ChIZ, Chelyabinsk, Russia). Analytical scales (VL-120M, Gosmetr, Saint-Petersburg, Russia) were used for mass determination.

### 2.3. Characterization Methods

#### 2.3.1. Sorption/Desorption Properties Characterization

Some of the most relevant hydrogel features are sorption capacity and swelling behavior in various medium. Water evaporation rate displays velocity of water desorption and ability to hold water molecules in the structure.

The properties were determined using the gravimetric method in distilled water (pH = 7) and phosphate-buffered salines (pH = 5.8, 7.4, 9.02). The buffer salines were prepared according to Russian Pharmacopoeia (13th edition). Square-shaped samples (dimensions—(10 × 10) ± 0.5 mm, thickness—0.12 ± 0.02 mm, mass—(0.10 ± 0.02 g) were placed in proper medium and left to freely swell. Then, samples were ejected from solution in certain time intervals and weighed. Excess water was removed via filter paper from surface of the samples.

Swelling value was calculated according to Equation (1):(1)$$Q=\frac{\left(m_{\max}-m_{0}\cdot \left(1-\gamma \right)\right)}{m_{0}\cdot \left(1-\gamma \right)}$$
where m_max_—mass of a swelled sample, g; m_0_—initial mass, g; γ—moisture content (entrapped water determined by TGA).

Analysis of desorption and sorption properties was carried out by diffusion Fick’s law (Equation (2)) and pseudosecond order equations [24] (Equation (3)).
(2)$$k\cdot t^{n}=Q_{t}/Q_{\max}$$
where Q_max_—maximum swelling value, g/g k_2_—constant of swelling velocity in pseudo second order equation, Q_t_—swelling at time, g/g, t—time, min.
(3)$$\frac{{\mathrm{dQ}}_{t}}{\mathrm{dt}}=k_{2}\cdot {\left(Q_{\max}-Q_{t}\right)}^{2}$$
where k—constant of swelling velocity in Fick’s law, n—diffusion degree.

Well-correlated results (R > 0.99) and good fitting with the lines were obtained by applying these equations to experimental data. Constants of these equations display velocity of swelling. Fick’s law was applied for the first period of swelling (until Q_t_ = 0.6 Q_max_), while pseudosecond order equation was applied for all periods of the research. The diffusion degree (n) shows type of diffusion.

#### 2.3.2. Crosslinking Density/Mesh Size Determination

Crosslinking density was calculated according to the Peppas–Barr–Howell equation [25] in water medium (pH = 7). Derived from the classical Flory–Rehner theory, this equation uses a macromolecular crosslinker in a significantly high concentration and produces more precise results than the original one.

Firstly, the volume fraction of the polymer after crosslinking in equilibrium with the swollen gel (φ_2_) was calculated (Equation (4)):(4)$$\varphi_{2}=\frac{\frac{1}{\rho_{2}}}{\frac{Q_{\max}}{\rho_{1}}+\frac{1}{\rho_{2}}}$$
where ρ_1_—water density, (1 g/cm^3^), ρ_2_—gelatin density, (1.35 g/cm^3^ [26]).

This allows us to determine the volume fraction of the solvent (φ_1_) (Equation (5)).
(5)$$\varphi_{1}=1-\varphi_{2}$$

The average molecular weight between crosslinks (M_c1_) represents average molecular weight of the primary polymer chains before crosslinking (Equation (6)).
(6)$$\frac{1}{M_{c1}}=\frac{2}{M_{n}}-\frac{\left(\frac{\upsilon }{V_{1}}\right)\cdot \left(\ln\left(1-\varphi_{2}\right)+\varphi_{2}+\chi \cdot \varphi_{2}\right)}{{\varphi_{2}}^{1/3}-\frac{\varphi_{2}}{2}}$$
where χ—the Flory−Huggins polymer−solvent interaction parameter (0.49 [27]), ν—the specific volume of the polymer (0.71 [28]), V_1_—molar volume of the swelling solvent, cm^3^/mole (18), M_n_—the average molecular weight of the linear polymer before crosslinking, g/mole (114.8 g/mole, calculated according to amino acid content in gelatin [29]).

The gel degree of crosslinking (X_n1_) is defined as ratio of crosslinks to repeating units (Equation (7)). This parameter represents crosslink density of network by numerical value.
(7)$$X_{n1}=\frac{\rho_{2}}{M_{c1}}$$

The mesh size (ε_1_) or the network correlation length defines the maximum molecule diameter, which can be entrapped in polymer network (Equation (8)).
(8)$$\varepsilon_{1}=\varphi_{2}^{-1/3}\cdot {\left(\mathrm{Cn}\cdot \eta \cdot \frac{M_{c1}}{M_{n}}\right)}^{1/2}\cdot l$$
where η—number of links per repeat unit (2 were used for calculation), l—the average length of a bond, A° (1.44 [30]), C_n_—Flory characteristic ratio (8.26 [30]).

#### 2.3.3. Gel Content Determination

Gel content of hydrogels (X_g_) was determined by drying them under NTP conditions, reaching equilibrium after the hydrogel samples swelled in water medium (Equation (9)). This number describes the total number of monomers, which take part in network formation.
(9)$$X_{g}=\frac{m_{a}}{m_{0}}\cdot 100$$
where m_a_—mass of a sample after drying, g.

#### 2.3.4. Porosity

The porosity of the hydrogels (X_p_) was determined by liquid displacement method [31] (Equation (10)). Briefly, the samples were weighed, and their volumes were measured. Then, the samples were immersed in ethanol until fully saturated and weighed. It displayed relative pore volume accessible for alcohol penetration.
(10)$$X_{p}=\frac{m_{e}-m_{0}}{\rho_{3}\cdot V_{2}}\cdot 100\%$$
where m_e_—mass of a sample after immersion in ethanol, g, ρ_3_—density of ethanol (0.79 [32]), g/cm^3^, V_2_—volume of a sample, cm^3^.

#### 2.3.5. Physico-Mechanical Analysis

Hydrogels need to correspond with operational conditions and have proper mechanical stability. Uniaxial stretch was used to find the tensile properties and approve applicability as functional material. Tensile strength and elongation at break were determined on a universal test frame (Instron 5966, Instron, Norwood, MA, USA) with following conditions: crosshead speed—10 mm/min, rectangular samples—30 × 8 × 8 mm). The samples were cut from equilibrium-swelled (water medium) films and placed with the initial distance of 10 mm between pneumatic tensile clamps. The hydrogels had insignificant deformation in the grip zone during mechanical experiments, and the destruction of the material was not visually defined. The geometry of the samples and experimental design was defined experimentally to provide best reproducibility.

For further calculation of hydrogel crosslinking density, the rubber elasticity theory was used [33]. First, linear region according to the elastic deformation theory force proportional to the deformation was found. The elastic modulus (G) can be computed from the slope of the line (Equation (11)):(11)$$F=G\cdot \left(\alpha -\frac{1}{\alpha^{2}}\right)$$
where F—the force applied to sample depending on its area, Pa, α—the elongation ratio, %.

The elastic modulus of an equilibrium-swelled hydrogel is inversely proportional to the molecular weight between crosslinks (M_c2_) (Equation (12)):(12)$$G=\frac{\rho_{2}}{M_{c2}}\cdot R\cdot T\cdot \varphi_{2}^{1/3}\cdot \varphi_{3}^{2/3}$$
where R—universal gas constant, J/(mole·K) (8.314), T—temperature, K (298), ϕ_3_—volume fraction of polymer after crosslinking before swelling.

The crosslinking density (X_n2_) can be obtained by Equation (13):(13)$$X_{n2}=\frac{\rho_{2}}{M_{c2}}$$

The mesh size (ε_2_) can be calculated by Equation (14):(14)$${ɛ}_{2}=\varphi_{2}^{1/3}\cdot {\left(C_{n}\cdot \frac{2M_{c2}}{M_{n}}\right)}^{1/2}\cdot l$$

#### 2.3.6. Study of the Thermal Properties

The thermal properties were obtained using a thermal analyser (SDT Q-600, TA instruments, New Castle, DE, USA) with the following conditions: nitrogen purge (40 mL/min), temperature 25 ÷ 600 °C, heating rates 5, 7.5, 10 K/min, mass of film samples 10 ± 2 mg, open alumina sample pans (100 μL). The samples were represented by disks cut from dried films.

Thermogravimetric analysis (TGA) was used for determination of thermal behavior. The temperatures of melting (T_m_) and glass transition (T_g_) were determined on differential scanning calorimetry (DSC) traces with 5 K/min heating rate as peak maximum and at midpoint, respectively. The initiation and the end of the step was determined by derivative thermogravimetry (DTG).

For further investigation of structure, the first step of thermal decomposition kinetics was studied. The extent of conversion (α) was determined as ratio of difference between initial mass and mass loss at a certain time to difference between initial mass and mass loss at the end of the step (Equation (15)).
(15)$$\alpha =\frac{m_{0}-m_{T}}{m_{0}-m_{f}}$$
where m_T_—mass at time, g, m_f_—mass at the end of the step, g.

The kinetic of thermal decomposition is usually described by Equation (16):(16)$$ẞ\frac{d\alpha }{\mathrm{dt}}=k\left(T\right)\cdot f\left(\alpha \right)$$
where β—heating rate, K/min, T—temperature, K, k(T)—reaction rate depended on T, f(α)—reaction model.

The rate constant (k(T)) obeys Arrhenius expression (Equation (17)):(17)$$k\left(T\right)=A\cdot \exp\left(-\frac{E_{a}}{\mathrm{RT}}\right)$$
where A—pre-exponential factor, E_a_—activation energy, kJ/mole, R—universal gas constant, J/(mole·K).

Equation (16) is usually transformed into the integral form (g(α)) for further calculations (Equation (18)).
(18)$$g\left(\alpha \right)=\int_{0}^{\alpha }\frac{d\alpha }{f\left(\alpha \right)}$$

Substituting the reaction model allows for the calculation of kinetic parameters from the slope of the plot of g(α) against T. The list of models used in this work (r) is presented in Table 1 [34].

There are several non-isothermal model-fitting methods that use approximations of Equation (18). The one used in this work is the Coats and Redfern (CR) method [35] (Equation (19)).
(19)$$\ln\frac{g\left(\alpha \right)}{T^{2}}=\ln(\frac{A\cdot R}{ẞ\cdot E_{a}}\left(1-\left(\frac{2\mathrm{RT}}{E_{a}}\right)\right)-\frac{E_{a}}{\mathrm{RT}}$$

The other is the Kennedy and Clark (KC) method [35] (Equation (20)). In this way, fitting the left side of Equations (19) and (20) to 1/T was used to determine E_a_ from the slope.
(20)$$\ln\left(\frac{ẞ\cdot g\left(\alpha \right)}{T-T_{0}}\right)=\ln\left(A\right)-\frac{E_{a}}{\mathrm{RT}}$$

Another widespread group of methods is isoconversional. These methods are based on the concept of the reaction model independence from the temperature. Two of models were used, the Kissinger–Akahira–Sunose (KAS, Equation (21)) and Flynn–Wall–Ozawa (FWO, Equation (22)) models, which provide ways to calculate the activation energies without the knowledge of a certain reaction mechanism [36].
(21)$$\ln\frac{ẞ}{T^{2}}=\ln\frac{A\cdot R}{E_{\alpha }\cdot g\left(\alpha \right)}-\frac{E_{\alpha }}{R\cdot T}$$
(22)$$\log ẞ=\log\frac{A\cdot E_{a}}{R\cdot g\left(\alpha \right)}-2.315-0.4567$$

#### 2.3.7. Water Vapor Transmission Rate

The test was carried out according to ASTM E96/E96M—10 [37]. The experiments were performed using a thermostat (CM 60-150/250-TBX, SM-Climate, Russia) at a temperature of (20 ± 1) °C and relative humidity of (40 ± 5)%. The water method was used. Firstly, polypropylene containers were filled with distilled water to mark 5 ± 1 mm from the neck. Hydrogel films (with a thickness of 0.12 ± 0.02 mm) were placed on the top of the containers with a surface effective area of 63.6 ± 0.05 cm^2^ and fixed on it. Edges of the films were sealed with foil to prevent oxygen transmittance bypass film. In addition, two control probes were made—one tightly closed with a cup and the other fully open. Then, samples were weighed with equipment and exposed in climate chamber with a controlled atmosphere for 24 h. After a defined time interval they were removed from the chamber, weighted and returned again. The water vapor transmission rate was calculated using Equation (23):(23)$$\mathrm{WVTR}=\frac{G}{t\cdot A}$$
where G/t—weight change by time (calculated from the slope of line), g/hour, A—test area, m^2^.

#### 2.3.8. Oxygen Permeability

Oxygen content in water samples was analyzed using a modified Winkler’s method [38]. Briefly, experiments were established as follows. The flat-bottomed flasks were filled with 200 mL of distilled water. The hydrogel films were placed on the top of the flasks with a test area of 9.6 ± 0.3 cm^2^ and sealed with foil to prevent oxygen penetration into water media. Two control probes without hydrogel samples (tightly closed and open flask) were used for comparison. Bottles were placed under NTP conditions for 24 h. After that all samples were removed, and the flasks were tightly closed to prevent further oxygen penetration. Then, using a syringe through a hole in a plug, 0.8 mL of manganese sulphate solution (50% *w*/*w*) and 2 mL of potassium iodide solution (15% *w*/*w* KI, 85% *w*/*w* KOH (10 M)) was added to each probe, preventing oxygen gaining from air. The flake-like brown precipitate (Mn(OH)_3_) appeared immediately.
4 Mn(OH)_2_ + O_2_ + 2 H_2_O → 4 Mn(OH_3_)

A total of 3.7 mL of sulphuric acid (50% *w*/*w*) was added for elementary iode forming. The precipitate dissolved, and dark red-violet solution was formed.
2 Mn(OH)_3_ + 6 H^+^ + 2 I^−^ → 2 Mn^2+^ + I_2_ + 6 H_2_O

Finally, the probe was titrated by sodium thiosulfate solution (0.5% *w*/*w*), using starch (0.1% *w*/*w*) as an indicator of equivalence point. This was determined by full discoloration of solution.
2 S_2_O_3_^2−^ + I_2_ → S_4_O_6_^2−^ + 2 I^−^

The oxygen content is quantified in terms of mole equivalent to iodine. 10 mL burette was used graduated every 0.05 mL.

#### 2.3.9. IR Spectroscopy Analysis

Fourier transform infrared (FT-IR) spectroscopy was applied to investigate obtained materials and display the differences of oxidant influence on chemical structures. IR spectras were found via spectrometer (Tensor 37, Bruker, Bremen, Germany) with an ATR attachment (diamond crystal, angle of incidence—45°, sample refractive index—1.4, number of reflections—3) in wavenumbers from 400 to 4000 cm^−1^ (resolution—2 cm^−1^, number of scans—1024). Data handling was carried out on Origin software v.9.6 (OriginLab, Northampton, MA, USA).

#### 2.3.10. Statistical Analysis

The results of experiments are displayed as arithmetic average value ± standard deviation of at least 3 samples used in the experiment. The *t*-test was used for comparing statistical significance of two groups. A value of *p* < 0.05 is considered statistically meaningful.

## 3. Results and Discussion

### 3.1. Performance Properties

#### 3.1.1. Sorption Studies

It is well known that hydrogels have a greater swelling capacity in clean water than in buffer saline [39]. This depends not only on the presence of salt ions in the water, but on the differences between the pH of various mediums as well. The experimental results show that the obtained materials follow the rule (Figure 2a).

The highest equivalent swelling values of all solutions were demonstrated when using hydrogen peroxide as an oxidizer. This might be an index of oxidizer efficacy in the formation of more polar and accessible structure for solution molecules. Moreover, there is clear dependence of swelling behavior on factors apart the oxidizer used. Swelling capacity decreased in the case of alternating the pH level from the neutral level, as well as when more salt was added. It was observed only for the samples obtained with hydrogen peroxide, except for DMSO-based hydrogels in acidic pH.

The analysis of sorption behavior in the early stage of swelling demonstrated the adherence of all samples in all mediums to Fick’s law [40] (n = (0.4 ÷ 0.6) ± 0.05). The rate of diffusion is much lower than the rate of relaxation and process controlled by diffusion. Additionally, the velocity of swelling is much less significant for the usage of small particle oxidizers (H_2_O_2_, DMSO) than large oxidizers (KMnO_4_, KIO_4_) (Figure 2b).

It should be mentioned that materials were more available in acid saline apart from the neutral and basic media for analysis in all time intervals (Figure 2c). The same-time, permanganate-based (compared with periodate hydrogels) sorption process slows down that moment. The gap in the diffusion rate among hydrogels obtained with small oxidizers is much less obvious compared to that of large molecules than in the first considered period.

It seems diffusion limits the relaxation of materials after 72 h, the rapid sorption of most formulation changes to prolonged sorption aiming to equilibrize swelling. The most accessible for solute penetration hydrogels were in acid pH media, which was approved by constants of both the pseudosecond order and Fick’s equation. That could be caused by the folding of gelatin molecules and providing solute easier access to negatively charged carboxylic groups [13]. Additionally, also of note is that in first stage of sorption, H-KIO_4_ and H kmnO_4_ demonstrated far higher velocity.

The previously highlighted behavior of hydrogels and sufficient sorption capacity might lead to a preference of their usage as pH-dependent materials [40].

#### 3.1.2. Water Evaporation Rate

The water evaporation rate displays the ability of polymer networks to retain water and exude it. The similar patterns of dehydration kinetic curves were observed in the case of the usage of various oxidizing systems (Figure 3).

Analysis of the first desorption period (≤0.6 m_t_/m_o_) showed that H-KIO_4_ had a significant mass loss in the first two hours (~30%), causing a major influence on Fick’s law degree value (Table 2). The gel content might have an impact on the first desorption period (Table 3), and a similar behavior was demonstrated by H-H_2_O_2_. Apart from that other samples were following an anomalous mechanism of diffusion (*n* > 0.5), as is characteristic for other hydrogels [41]. The most available material for water diffusion from the inside of the hydrogels was H-H_2_O_2_, while the least available was H-DMSO.

#### 3.1.3. Gel Content/Porosity Analysis

The gel content of obtained samples was significantly high, in the range of (74.3 ÷ 95.9) ± 0.4% (Table 3). This demonstrates that most gelatins take part in crosslinking reactions. The porosity measurements display that the total value of pores was strongly dependent on the used oxidizing system. The desorption properties of H-KIO_4_ and H kmnO_4_ were the most rapid and the least rapid among all recipes, while their porosity followed an opposite trend.

It must be highlighted that KIO_4_ was less effective as an oxidant for studying synthesis than KMnO_4_. Hydrogels obtained with usage of potassium periodate had the lowest gel content values. This might be caused by the large ionic radius of iodate ions, which causes some issues with delivering ions directly to target groups. The ionic radius for all oxidants particles is O^2−^—1.58 A°, (CH_3_)_2_SO—1.83 A°, IO_4_^−^—2.31 A°, MnO_4_^−^—2.2 A° [42]. Likewise, the most effective was hydrogen peroxide at that point. This was confirmed by crosslinking density calculations from sorption studies but not those of physico-mechanical studies.

#### 3.1.4. Water Vapor Transmission Rate/Oxygen Permeability

The water vapor transmission rate shows the time-dependent mass difference through the material unit of the area and thickness induced by unit vapor pressure difference. In control probes that were open (3046 ± 86) and closed (50 ± 1) g/(h·m^2^). Regarding water vapor, the most permeable samples were obtained by using hydrogen peroxide as an oxidizer, and the least permeable samples were determined to be those obtained using potassium periodate (Table 4).

Oxygen permeability displays the concentration of dissolved oxygen in water. The control probes results were (7.19 ± 0.03) and (9.68 ± 0.05) mg/L (for closed and open probes, respectively). This opinion was formed through a comparison of the results. The oxygen content in water usually ranges from 0 to 15 mg/L [43]. The least permeable was hydrogen peroxide, while the most permeable was potassium periodate in synthesized samples.

The results correlate with values from other studies proposing wound dressing application [44,45].

#### 3.1.5. Physico-Mechanical Tests

Fully hydrated hydrogels display sufficiently high tensile properties to find appropriate application (Figure 4). The tensile strength of the hydrogels was higher for H kmnO_4_ and H-KIO_4_ than for H-H_2_O_2_ and H-DMSO. All of the considered hydrogels display typical elastomer behavior. The value of elongation at break varied from 150 to 300%, depending on the oxidizer used. The obtained results show quite similar values for fully hydrated hydrogels [46,47].

### 3.2. Structure Characterisation

#### 3.2.1. Thermal Analysis

The TG and DTG curves obtained for different heating rates are presented in Figure 5. All curves for different heating rates correlated with one type samples and shifted to higher temperatures with a rising heating rate. Three steps can be seen and related to the evaporation of bounded water (25 ÷ 250) °C—I, degradation of the gelatin (250 ÷ 350) °C—II and decomposition of the residues (450 ÷ 550) °C—III, similar to native gelatin [48].

The weight loss at initial (T_i_), peak (T_p_) and final (T_f_) decomposition temperatures are listed in Table 5 for a 10 K/min heating rate according to the proper step. This clearly shows that H kmnO_4_ and H-KIO_4_ have higher peak temperatures in step II and lower ones in step III in comparison with H-H_2_O_2_ and H-DMSO. The weight residues at the end of step II and III also indicate greater thermal stability of the hydrogels obtained with large oxidants.

The DSC traces have a broad endothermic peak (30 ÷ 200) °C, attributed to evaporation of water from the structure. Additionally, DSC curves have exothermic peaks at temperatures of (390 ÷ 590) °C, corresponding to the polymer decomposition, and a peak at about (305 ÷ 320) °C, apparently responsible for melting (Figure 6). T_g_ was determined by a step on the baseline, which appeared because of different heat capacities below and above. The results here showing the character of thermal transitions are relatively similar to those of other studies [49,50].

The T_g_ and T_m_ of the samples are presented in Table 5. Apart from relatively close values of temperatures of other samples two of them differ sharply: T_g_ of H-DMSO and T_m_ of H-KIO_4_ (Table 6).

Firstly, the non-isothermal kinetics of stage I was determined by isoconversional methods to obtain the dependences of activation energy from the extent of conversion (Figure 7). Step I was chosen for kinetic analysis as it demonstrates the easiness of the water molecules’ detachment. Indirectly, it may indicate the relative difference of structure properties. The energy distribution in the conversion range of 0.2 ÷ 0.8 is very low for each hydrogel sample, and the curves follow the same trend of slow linear increasing in extent for two considered isoconversional methods. This demonstrates that the reactions obey the Arrhenius equation and follows the single-step reaction mechanism [51]. Apart from H kmnO_4_, which demonstrates rapid growth (α ≥ 0.6). This could be an indication of an earlier start of step II.

The values obtained by FWO and KAS differ much in the conversion range of 0.2 ÷ 0.6 (about two times). The calculated means for that span tend to increase from 4 ÷ 18 to 14 ÷ 38 kJ/mol depending on the recipe and method used. These low values are characteristic of the reversible exothermal stage and correlate with another work [52].

After applying the isoconversional methods the model fitting methods were used in the linear region (0.2 ≤ α ≤ 0.6) for the calculation energy of activation (Table 7). It was expected to follow the single-step reaction mechanism, so the reason was to establish which reaction model (Table 1) gives the best coefficient of correlation. After fitting models to TGA data one model (*r*) was chosen for each heating rate and recipe.

The values of the average activation energy were in good agreement with those obtained by model fitting methods. Only by applying Coats–Redfern method for H-KIO_4_ was significant variation found with Kennedy–Clark and isoconversional methods [36]. According to that H-KIO_4_ had the lowest activation energy for the three calculation methods. With increasing heating rate activation energy also increases. Most of the models applied for the CR method were well correlated with three-dimensional diffusion models (18 and 19). The values obtained by the CR method were greater than those obtained by the KC method [34].

#### 3.2.2. Crosslinking Density

The crosslinking density and mesh size evaluated by the sorption method demonstrate narrow distribution in the case of the usage of different oxidizing systems (Table 8). However, analysis through the rubber elasticity theory clearly shows that the more significantly crosslinked network was synthesized with more powerful oxidizers, such as potassium permanganate and periodate.

The differences between crosslinking density values obtained from sorption and physico-mechanical data can be explained by the limitations of the rubber elasticity theory. The rubber elasticity theory, similarly to the Flory–Rehner theory of polymer network swelling, used an approach of following narrow Gaussian distribution by polymer network. The hydrogels synthesized with bimodal chain length distribution create a more durable network [53]. Moreover, the rubber elasticity theory must consider factors such as chain length and flexibility of the network. This strongly affects the values of crosslinking density. The bulky sidechains and chain complexity of a polymer network contributes to entropic chain energy. This provides more deviation from the ideal network model proposed for uniaxial isotropic elasticity [54], while sorption obtained crosslinking density values takes into account for both elastically deformable and nondeformable nature of crosslinked segments.

#### 3.2.3. IR Spectroscopy Analysis

Intensive peaks around 1654, 1552 and 1238 cm^−1^ were attributed to amide I, II and III exactly, which are widespread in peptides (Figure 8). The high band around 1654 cm^−1^ corresponds with C=O valent vibration connected with N-H deformation oscillation in an amide group. Amide II is responsible for the deformation oscillation of N–H correlated with valent vibrations of C–N. Amide I and II bands of hydrogels have less wavenumbers than those in native gelatin. This may be the cause of binding tannin and gelatin. Moreover, the spectral band of Schiff’s base matches amide I. Differences of peak position at about 1631 cm^−1^ might be a consequence of the gelatin and tannin reaction by the Schiff base formation mechanism.

The bands of carbon bonds (–CH_3_ and –CH_2_) with 2922 and 2855 cm^−1^ wavenumbers have more intensity in hydrogels than in native gelatin. The same trend is observed for carboxylic groups at around 1740 cm^−1^.

The intensity of the absorption band around 1450 cm^−1^ shows an increase with the crosslinking of hydrogels in addition to the above-mentioned peaks. This band is associated with CH=N group vibrations [55]. The alteration of color also follows the same tendency. This occurs due to brighter native gelatin and hydrogels obtained with the usage of DMSO and H_2_O_2_ and darker hydrogels obtained by using KMnO_4_ and KIO_4_.

Increasing intensity of absorption near 1027 cm^−1^ (C–OH) indicates the existence of tannin in hydrogels. The formation of new peaks around 928 and 865 cm^−1^ illustrate the presence of –CH from tannin benzene rings in crosslinked hydrogels. This might be caused by the establishment of new bonds between gelatin amino groups and tannin molecules [12].

Another argument for the amino groups’ reaction with tannin is wavenumber shift of amid A band from 3320 (for native gelatin) to 3290 cm^−1^ (for crosslinked hydrogels).

The second derivations of IR spectras were gathered to show more clear differences between native gelatin and hydrogels (Figure 9). All newly formed peaks are contrasted by indication of exact wavenumbers. All three characteristic bands for the secondary structure exist near sensitive structure alterations amide I band (1690–1600 cm^−1^) and have shifts by wavenumbers in the comparison to native gelatin. Moreover, there are notable formations of new absorption bands around 1578 and 1523–1511 cm^−1^ [16]. The first may indicate amino groups changing, while the second might be the appearance of NH^3+^.

The occurrence of new bands around 1467, 1377 and 1366 cm^−1^ (typical for deformation vibrations of C–H groups) might be caused by a benzene ring embedding in the hydrogel structure. The bands near 1027 and 990 cm^−1^ demonstrate the inclusion of tannin in the hydrogel structure [21].

## 4. Conclusions

The usage of the oxidizing systems for synthesizing gelatin–tannin hydrogels was performed. The study shows the possibility of using hydrogen peroxide, dimethylsulfoxide, potassium permanganate and periodate in that role. The influence of oxidants on functional and structure properties was significant. The forming of hydrogel structures was confirmed by IR-spectroscopy. The hydrogels have water/oxygen permeability and fine mechanical properties appropriate for wound dressing application. The crosslinking density of materials supports the stiffness of hydrogels, obtained by using small oxidants (H-H_2_O_2_, H-DMSO). Thermal properties and decomposition represent that the variation of structure changes with the type of oxidizing system. Kinetic of water residues evaporation clearly shows that materials allow free diffusion of water molecules in all dimensions. The pH-dependent behavior in acid media make them suitable for drug delivery application. The most perspective oxidizing system was that based on hydrogen peroxide regarding the functional properties of obtained materials. These materials are still under investigation for synthesizing scaffolds in order to discover future useful medicine applications.

## Figures and Tables

**Figure 1 polymers-14-00150-f001:**
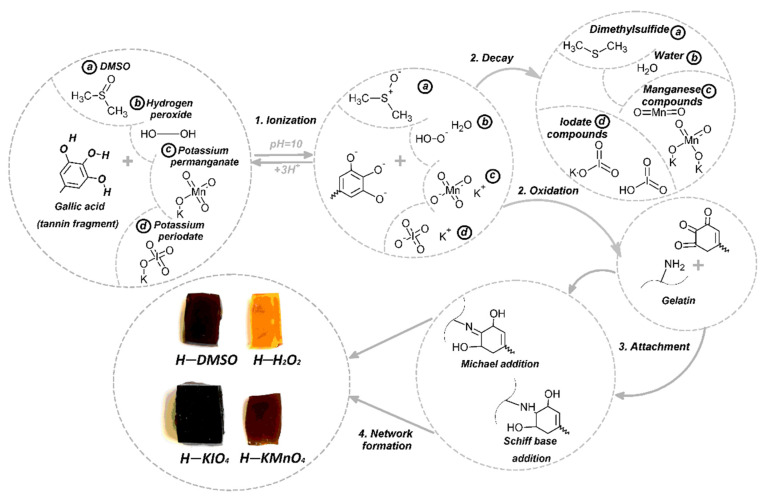
Scheme of the reactions.

**Figure 2 polymers-14-00150-f002:**
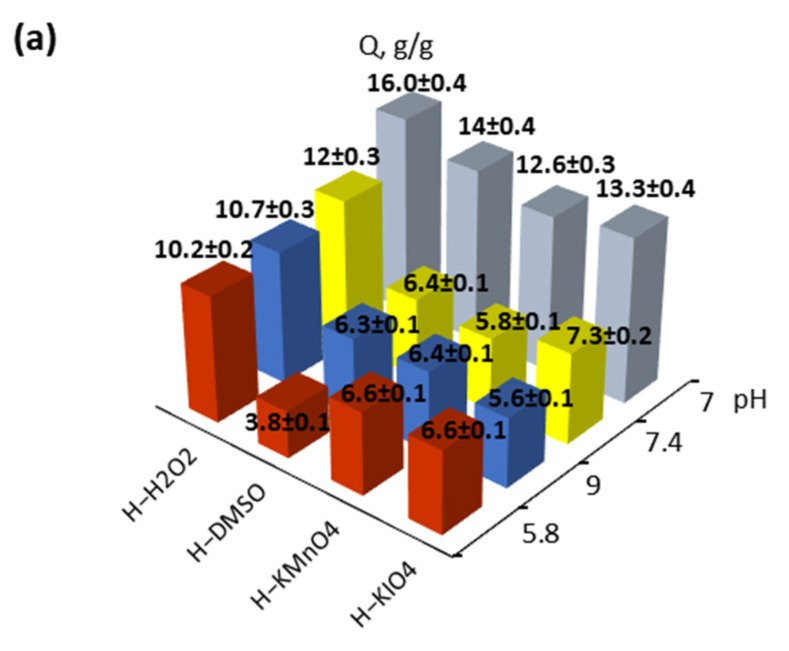
Sorption properties of hydrogels depending on medium pH and type of oxidizer: (**a**) swelling degree after 72 h; (**b**) values of constants in Fick’s law; (**c**) values of constants in pseudosecond order equation.

**Figure 3 polymers-14-00150-f003:**
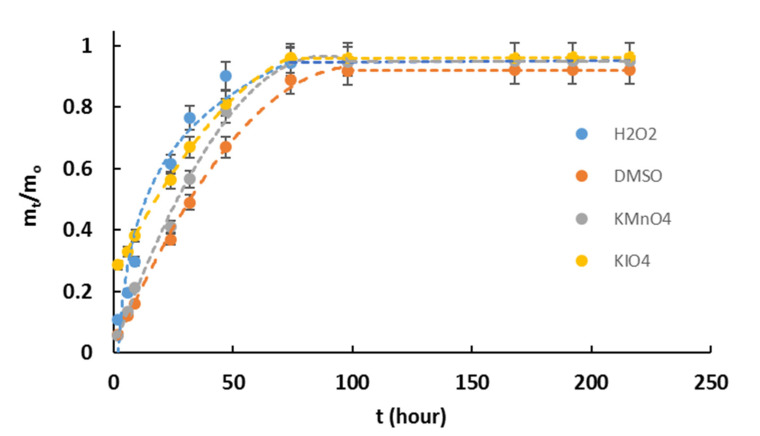
Graphs of changing water content in hydrogels during time of exponation.

**Figure 4 polymers-14-00150-f004:**
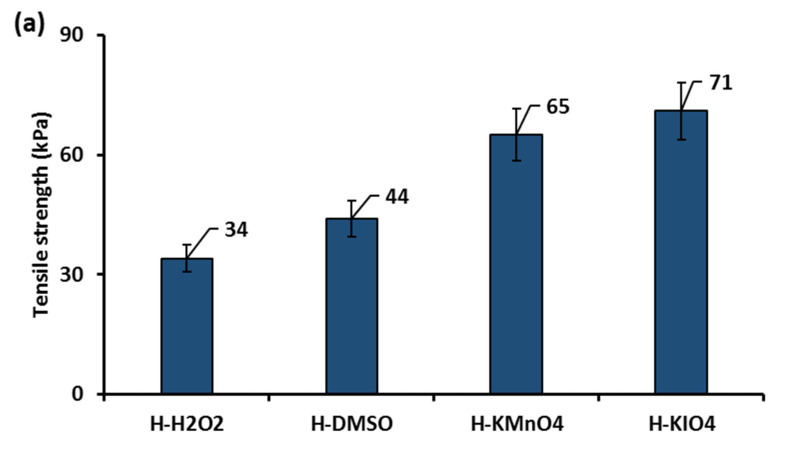
Physico-mechanical properties of hydrogels: (**a**) tensile strength; (**b**) elongation at break.

**Figure 5 polymers-14-00150-f005:**
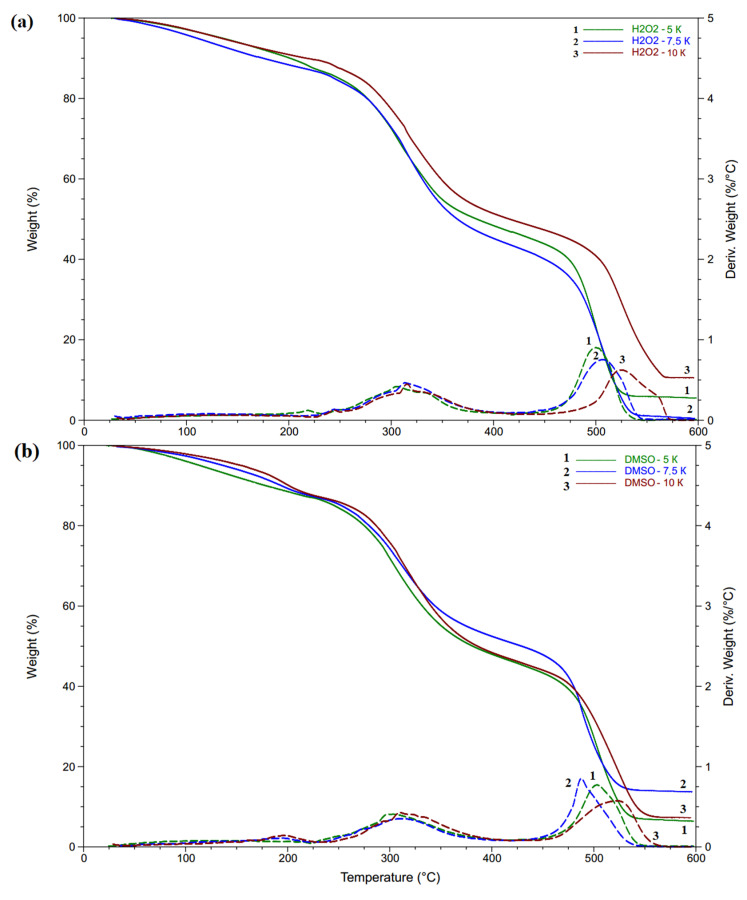
TGA and DTG curves of the hydrogels, obtained with various heating rates: (**a**) H-H_2_O_2_; (**b**) H-DMSO; (**c**) H kmnO_4_; (**d**) H-KIO_4_ (1—5 K/min, 2—7.5 K/min, 3—10 K/min, solid line—TGA, dashed line—DTG).

**Figure 6 polymers-14-00150-f006:**
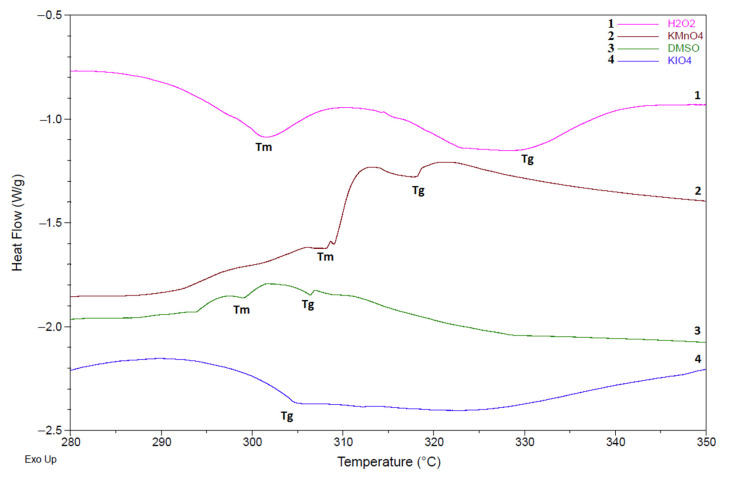
Determination of T_g_ and T_m_ on DSC trace of samples (1—H-H_2_O_2_; 2—H kmnO_4_; 3—H-DMSO; 4—H-KIO_4_).

**Figure 7 polymers-14-00150-f007:**
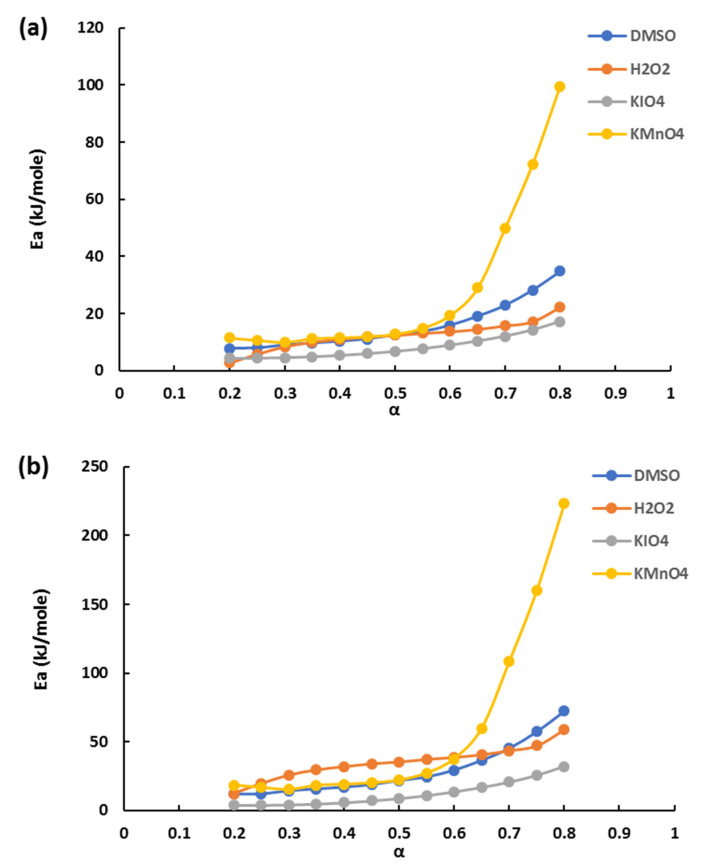
Energy of activation for each conversion step (0.2 ÷ 0.8) using (**a**) FWO and (**b**) KAS methods.

**Figure 8 polymers-14-00150-f008:**
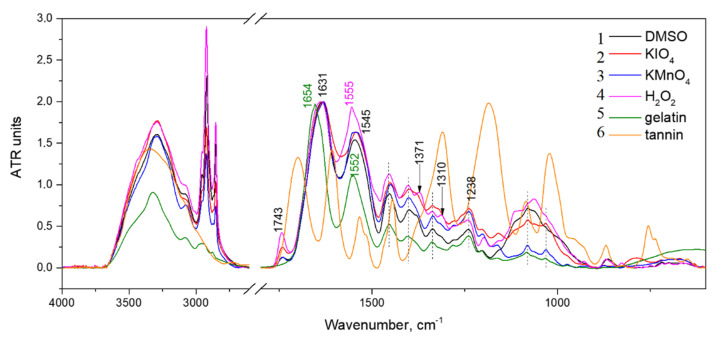
IR spectras of hydrogels, obtained by using 1—DMSO, 2—KIO_4_, 3—KMnO_4_, 4—H_2_O_2_, and spectras of 5—native gelatin and 6—tannin.

**Figure 9 polymers-14-00150-f009:**
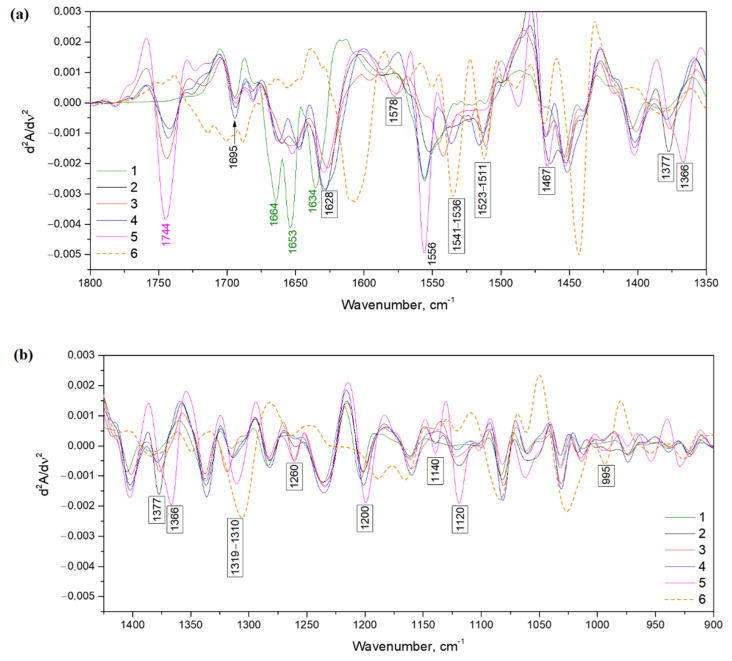
Second derivations of IR spectras: (**a**) 1800 ÷ 1350 cm^−1^, (**b**) 1400 ÷ 900 cm^−1^; 1—gelatin and hydrogels obtained by usage different oxidizers: 2—DMSO, 3—KIO_4_, 4—KMnO_4_, 5—H_2_O_2_ and 6 –tannin.

**Table 1 polymers-14-00150-t001:** Reaction models.

No	Reaction Model (*r*)	f(α)	g(α)
1	Power law	4α^0.75^	α^0.25^
2	Power law	3α^0.66^	α^0.33^
3	Power law	2α^0.5^	α^0.5^
4	Power law	0.66α^−0.5^	α^1.5^
5	Zero-order (Polany–Winger equation)	1	α
6	Phase boundary-controlled reaction (contracting area, i.e., bidimensional shape)	2(1 − α)^0.5^	1 − (1 − α)^0.5^
7	Phase boundary-controlled reaction (contracting area, i.e., tridimensional shape)	3(1 − α)^0.66^	1 − (1 − α)^0.33^
8	First-order (Mampel)	(1 − α)	−ln(1 − α)
9	Three-halves order	(1 − α)^1.5^	2((1 − α)^−0.5^ − 1)
10	Second-order	(1 − α)^2^	(1 − α)^−1^ − 1
11	Third-order	(1 − α)^3^	0.5((1 − α^)−2^ − 1)
12	Avrami–Erofeev (n = 1.5)	1.5(1 − α)(−ln(1 − α))^0.33^	−ln(1 − α)^0.66^
13	Avrami–Erofeev (n = 2)	2(1 − α)(−ln(1 − α))^0.5^	−ln(1 − α)^0.5^
14	Avrami–Erofeev (n = 3)	3(1 − α)(−ln(1 − α))^0.66^	−ln(1 − α)^0.33^
15	Avrami–Erofeev (n = 4)	4(1 − α)(−ln(1 − α))^0.75^	−ln(1 − α)^0.25^
16	One-dimensional diffusion	0.5α	α^2^
17	Two-dimensional diffusion (bidimensional shape), Valensi equation	1/(−ln(1 − α))	(1 − α)ln(1 − α) + α
18	Three-dimensional diffusion (tridimensional shape), Jander equation	3(1 − α)^0.33^/(2(1 − α)^−0.33^ − 1)	(1 − (1 − α)^0.33^)^2^
19	Three-dimensional diffusion (tridimensional shape), Ginstling–Brounstein equation	1.5((1 − α)^−0.33^ − 1)	(1 − 0.66α) − (1 − α)^0.66^

**Table 2 polymers-14-00150-t002:** Analysis results of water evaporation rate by Fick’s law.

Sample	*k* (g/(mmole·min))	*n*
H-H_2_O_2_	0.06 ± 0.01	0.71 ± 0.03
H-DMSO	0.03 ± 0.01	0.77 ± 0.03
H-KMnO_4_	0.03 ± 0.01	0.82 ± 0.04
H-KIO_4_	0.22 ± 0.03	0.27 ± 0.02

**Table 3 polymers-14-00150-t003:** Hydrogel properties, determined by sorption studies.

Sample	X_g_ (%)	X_p_ (%)
H-H_2_O_2_	95.9 ± 0.4	37.5 ± 0.4
H-DMSO	86.7 ± 0.3	56.1 ± 0.5
H-KMnO_4_	93.2 ± 0.4	80.8 ± 0.6
H-KIO_4_	74.3 ± 0.3	17.3 ± 0.2

**Table 4 polymers-14-00150-t004:** Results of permeability properties of polymer films.

Sample	WWTR, (g/(h·m^2^))	OP (mg/L)
H-H_2_O_2_	1848 ± 54	7.82 ± 0.04
H-DMSO	1729 ± 50	8.94 ± 0.04
H-KMnO_4_	1793 ± 52	9.29 ± 0.05
H-KIO_4_	1542 ± 45	8.38 ± 0.05

**Table 5 polymers-14-00150-t005:** TGA results.

Sample	Step	T_i_ (°C)	T_p_ (°C)	T_f_ (°C)	Weight at T_f_ (%)
H-H_2_O_2_	I	37 ± 1	133 ± 1	223 ± 1	89.7 ± 0.2
	II	227 ± 1	316 ± 1	424 ± 1	48.6 ± 0.1
	III	448 ± 1	525 ± 1	579 ± 1	10.6 ± 0.1
H-DMSO	I	37 ± 1	197 ± 1	232 ± 1	88.6 ± 0.2
	II	240 ± 1	311 ± 1	418 ± 1	47.9 ± 0.1
	III	443 ± 1	523 ± 1	578 ± 1	7.3 ± 0.1
H-KMnO_4_	I	39 ± 1	184 ± 1	232 ± 1	87.1 ± 0.2
	II	236 ± 1	325 ± 1	391 ± 1	61.9 ± 0.2
	III	396 ± 1	514 ± 1	590 ± 1	36.5 ± 0.1
H-KIO_4_	I	38 ± 1	159 ± 1	220 ± 1	88.2 ± 0.2
	II	225 ± 1	324 ± 1	402 ± 1	63.2 ± 0.2
	III	435 ± 1	500 ± 1	566 ± 1	33.2 ± 0.1

**Table 6 polymers-14-00150-t006:** Thermal properties of hydrogels.

Sample	T_m_ (°C)	T_g_ (°C)
H-H_2_O_2_	319 ± 1	301 ± 1
H-DMSO	306 ± 1	300 ± 1
H-KMnO_4_	318 ± 1	303 ± 1
H-KIO_4_	312 ± 1	261 ± 1

**Table 7 polymers-14-00150-t007:** Comparison of activation energy obtained by model-fitting methods.

Heating Rate (K/min)		Coats-Redfern Method				Kennedy-Clark Method			
		H-H_2_O_2_	H-DMSO	H-KMnO_4_	H-KIO_4_	H-H_2_O_2_	H-DMSO	H-KMnO_4_	H-KIO_4_
	*r*	17	18	18	19	19	7	16	7
*5*	*R* ^2^	0.9999	0.9998	1.0000	0.9999	0.9997	0.9998	0.9976	0.9999
	*E_α_*	33.1	39.2	34.2	35.4	30.5	24.7	30.5	16.4
	*r*	18	18	18	18	18	16	5	18
*7.5*	*R* ^2^	0.9994	0.9998	0.9997	0.9993	0.9998	0.9997	0.9976	0.9996
	*E_α_*	37.6	38.4	40.7	36.8	31.6	28.9	28.1	25.1
	*r*	18	18	18	18	19	16	18	12
*10*	*R* ^2^	0.9992	0.9994	1.0000	0.9994	0.9986	0.9997	0.9917	0.9981
	*E_α_*	41.8	40.6	40.8	45.3	31.9	30.7	32.3	26.7

**Table 8 polymers-14-00150-t008:** Comparison of hydrogel crosslinking density obtained by various methods.

Sample	Sorption Analysis			Physico-Mechanical Analysis		
	Mc_1_ (g/mol)	X_n1_·10^3^ (mol/cm^3^)	ε_1_ (A°)	Mc_2_ (g/mol)	X_n2_·10^3^ (mol/cm^3^)	ε_2_ (A°)
H-H_2_O_2_	66.7	20.24	12.5	226.4	5.96	23.4
H-DMSO	67.7	19.95	12.1	197.0	6.85	20.9
H-KMnO_4_	68.5	19.70	11.7	80.0	16.88	12.9
H-KIO_4_	68.1	19.83	11.9	63.5	21.27	11.7

## Data Availability

The authors confirm that the data supporting the findings of this study are available within the article.

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
