# Peer review of "The Influence of Oxidant on Gelatin–Tannin Hydrogel Properties and Structure for Potential Biomedical Application"

_polymers, 2021, doi:10.3390/polym14010150_

Round 1

Reviewer 1 Report

Paper ID: polymers-1529396

The influence of oxidant on gelatin-tannin hydrogel properties and structure for potential biomedical application   Konstantin Olegovich Osetrov * , Mayya Valeryevna Uspenskaya , Vera Evgenevna Sitnikova  

Reviewer’s comments:

Some very small corrections:

  1. In point 2.2.5 Please explain how the hydrogel samples were mounted in the holders of the testing machine. Was their shape distorted in the jaws?
  2. In point 2.2.9 please add information on the measurement method based on ATR. 

Conclusion: Publish after minor corrections

Author Response

The authors express their gratitude to the reviewers for the great work done during the review of the article, the suggestions made and questions asked. The article has been majorly revised. The responses has been written to all comments (except for grammatical, lexical, and stylistic errors, which have been corrected taking into account the comments of reviewers). We hope that the corrected version of the article meets all the requirements necessary for publication in the journal "Polymers".

The following are comments with responses to comments

Reviewer’s comments:

Some very small corrections:

  1. In point 2.2.5 Please explain how the hydrogel samples were mounted in the holders of the testing machine. Was their shape distorted in the jaws?
  2. In point 2.2.9 please add information on the measurement method based on ATR. 

Conclusion: Publish after minor corrections

Responses:

  1. The hydrogels were mounted in pneumatic tensile grips with insignificant deformation in grips zone during mechanical experiments. The destruction of the material wasn’t visually defined.

The corrections have been made in the manuscript.

  1. The ATR method info was added to the text included following: diamond crystal, angle of incidence – 450, sample refractive index – 1.4, number of reflections – 3.

The corrections have been made in the manuscript.

Reviewer 2 Report

The work " The Influence of Oxidant on Gelatin-tannin Hydrogel Proper-2 ties and Structure for Potential Biomedical Application " presents a representative analysis of the proposed study, and I recommend publication after the following corrections:

  1. Introduction:

The section introduction was have been good writing and need minor corrections:

#Line 27: Remove bold from the sentence.

#Line 88: Insert full stop in the sentence.

  1. Materials and Methods

I suggest that separating subsection 2.1 in:

2.1 Material

2.2 Synthesis

# Line 96 to 108: How the parameters of solution formation was defined?

#Line 167 to 172: What method / pattern used to perform the analysis?

#Line 183: Remove the space before the beginning of the sentence.

#Line 261: Insert information about the equipment (brand, country). Verify all the manuscript.

#Line 266: Remove the double "in" in the sentence.

  1. Results and discussion

#Line 303 to 304: What is the reference to the statement?

#Line 313 to 317: Is there some explanation for this behavior? What can be attributed to the mass loss in the first two hours (~ 30%)?

#Line 412: The authors mentioned a "significant variation" between the Coats-Redfern and Kennedy-Clark methods. Was any test performed to show whether the differences were statistically significant?

The “results and discussion” section needs to be revised. The results are clearly displayed. However, the discussion needs to be improved and I strongly recommend inserting more references that support the statements made as well as comparing the results obtained with results from the literature. Besides that, it would be interesting to explain how the results impact future application.

  1. Conclusions

In line with that suggest previously, I recommend inserting a link between the obtained results and the final application of this product.

Author Response

The authors express their gratitude to the reviewers for the great work done during the review of the article, the suggestions made and questions asked. The article has been majorly revised. The responses has been written to all comments (except for grammatical, lexical, and stylistic errors, which have been corrected taking into account the comments of reviewers). We hope that the corrected version of the article meets all the requirements necessary for publication in the journal "Polymers".

The following are comments with responses to comments:

Reviewer’s 2 comments:

The work " The Influence of Oxidant on Gelatin-tannin Hydrogel Proper-2 ties and Structure for Potential Biomedical Application " presents a representative analysis of the proposed study, and I recommend publication after the following corrections:

  1. Introduction:

The section introduction was have been good writing and need minor corrections:

#Line 27: Remove bold from the sentence.

#Line 88: Insert full stop in the sentence.

  1. Materials and Methods

I suggest that separating subsection 2.1 in:

2.1 Material

2.2 Synthesis

# Line 96 to 108: How the parameters of solution formation was defined?

#Line 167 to 172: What method / pattern used to perform the analysis?

#Line 183: Remove the space before the beginning of the sentence.

#Line 261: Insert information about the equipment (brand, country). Verify all the manuscript.

#Line 266: Remove the double "in" in the sentence.

  1. Results and discussion

#Line 303 to 304: What is the reference to the statement?

#Line 313 to 317: Is there some explanation for this behavior? What can be attributed to the mass loss in the first two hours (~ 30%)?

#Line 412: The authors mentioned a "significant variation" between the Coats-Redfern and Kennedy-Clark methods. Was any test performed to show whether the differences were statistically significant?

The “results and discussion” section needs to be revised. The results are clearly displayed. However, the discussion needs to be improved and I strongly recommend inserting more references that support the statements made as well as comparing the results obtained with results from the literature. Besides that, it would be interesting to explain how the results impact future application.

  1. Conclusions

In line with that suggest previously, I recommend inserting a link between the obtained results and the final application of this product.

Responses:

  1. The corrections have been made in the introduction of the manuscript.
  2. The separation of subsection 2.1 was done.

# Line 96 to 108: The synthesis parameters was defined previously (unpublished data).

#Line 167 to 172: It was problematic to choose one of the standard methods due to high water content of hydrogel materials. The geometry of the samples and experimental design was defined experimentally to provide best reproducibility.

The corrections have been made in the manuscript.

#Line 183: The correction has been made in the manuscript.

#Line 261: The brand and country information added to all equipment.

#Line 266: The sentence was corrected.

  1. #Line 303 to 304: What is the reference to the statement?

The reference was added (13.   Sorushanova, A.; Delgado, L.M.; Wu, Z.; Shologu, N.; Kshirsagar, A.; Raghunath, R.; Mullen, A.M.; Bayon, Y.; Pandit, A.; Raghunath, M.; et al. The Collagen Suprafamily: From Biosynthesis to Advanced Biomaterial Development. Adv. Mater. 2019, 31, 1–39, doi:10.1002/adma.201801651.).

#Line 313 to 317: It might be caused by the lowest gel content of hydrogels (table 3), close behavior was demonstrated by H-H2O2

The correction has been made in the manuscript.

#Line 412: That is widely known effect related to the limitations of calculation methods. The statistical analysis wasn’t applied for comparing computed values.

The reference added to the text.

The “results and discussion” section was revised: added more references for comparison with close works, expanded discussion (subsections 3.1.1, 3.1.2, 3.1.4, 3.1.5, 3.2.1)

  1. The correction has been made in the manuscript.

Reviewer 3 Report

The manuscript “The Influence of Oxidant on Gelatin-tannin Hydrogel Properties and Structure for Potential Biomedical Application” deals with the production of gelatin-tannin hydrogels using some oxidants. Several characterizations were performed on the obtained samples, such as sorption capacity, tensile strength, and permeability for water/oxygen, that indicated these materials as suitable for the biomedical field. This is a good and accurate work, perfectly in line with the Journal aims. Therefore, the publication is recommended; but after some revisions, as described below:

- Abstract. Add quantitative results to this section.

- Introduction. The state of the art related to the production of hydrogels/aerogels can be enlarged adding some recent and innovative works in the field, as the works of Baldino et al., Smirnova et al., etc…

- Use “.” instead of “,” as decimal separator.

- Check subscripts in the various formula.

- Compare the obtained results with the previous literature to underline the advantages and the performance of the produced gels.

- Follow the journal template.

- Typos are present. Check and correct them.

Author Response

The authors express their gratitude to the reviewers for the great work done during the review of the article, the suggestions made and questions asked. The article has been majorly revised. The responses has been written to all comments (except for grammatical, lexical, and stylistic errors, which have been corrected taking into account the comments of reviewers). We hope that the corrected version of the article meets all the requirements necessary for publication in the journal "Polymers".

The following are comments with responses to comments:

Reviewer’s 3 comments:

The manuscript “The Influence of Oxidant on Gelatin-tannin Hydrogel Properties and Structure for Potential Biomedical Application” deals with the production of gelatin-tannin hydrogels using some oxidants. Several characterizations were performed on the obtained samples, such as sorption capacity, tensile strength, and permeability for water/oxygen, that indicated these materials as suitable for the biomedical field. This is a good and accurate work, perfectly in line with the Journal aims. Therefore, the publication is recommended; but after some revisions, as described below:

- Abstract. Add quantitative results to this section.

- Introduction. The state of the art related to the production of hydrogels/aerogels can be enlarged adding some recent and innovative works in the field, as the works of Baldino et al., Smirnova et al., etc…

- Use “.” instead of “,” as decimal separator.

- Check subscripts in the various formula.

- Compare the obtained results with the previous literature to underline the advantages and the performance of the produced gels.

- Follow the journal template.

- Typos are present. Check and correct them.

Responses:

- Use “.” instead of “,” as decimal separator.

- Check subscripts in the various formula.

- Follow the journal template.

- Typos are present. Check and correct them.

The corrections have been made in the manuscript.

- Abstract. The quantitative results was supplemented to the section.

- Introduction. The proposed works present high-quality researches in aerogels field, technology and their biomedical application. However, inserting them in current work may mislead potential reader because of a slightly different topic.

The “results and discussion” section was revised: added more references for comparison with close works, expanded discussion (subsections 3.1.1, 3.1.2, 3.1.4, 3.1.5, 3.2.1)

Round 2

Reviewer 3 Report

The authors performed the modifications proposed by the Reviewer and improved the manuscript.